# Infection of Brain Organoids and 2D Cortical Neurons with SARS-CoV-2 Pseudovirus

**DOI:** 10.3390/v12091004

**Published:** 2020-09-08

**Authors:** Sang Ah Yi, Ki Hong Nam, Jihye Yun, Dongmin Gim, Daeho Joe, Yong Ho Kim, Han-Joo Kim, Jeung-Whan Han, Jaecheol Lee

**Affiliations:** 1Epigenome Dynamics Control Research Center, School of Pharmacy, Sungkyunkwan University, Suwon 16419, Korea; nam6422@hanmail.net (K.H.N.); wlgpp405@gmail.com (J.Y.); kdmin3524@naver.com (D.G.); daeho1027@g.skku.edu (D.J.); jhhan551@skku.edu (J.-W.H.); 2SKKU Advanced Institute of Nanotechnology (SAINT), Sungkyunkwan University, Suwon 16419, Korea; yonghokim94@gmail.com; 3Department of Biomedical Engineering, Sungkyunkwan University, Suwon 16419, Korea; 4Department of Nano Engineering, Sungkyunkwan University, Suwon 16419, Korea; 5Biomedical Institute for Convergence at SKKU (BICS), Sungkyunkwan University, Suwon 16419, Korea; 6Imnewrun Biosciences Inc., Suwon 16419, Korea; han-joo.kim@imnewrun.com

**Keywords:** SARS-CoV-2, pseudovirus, brain organoid, cortical neuron, ACE2

## Abstract

Since the global outbreak of SARS-CoV-2 (COVID-19), infections of diverse human organs along with multiple symptoms continue to be reported. However, the susceptibility of the brain to SARS-CoV-2, and the mechanisms underlying neurological infection are still elusive. Here, we utilized human embryonic stem cell-derived brain organoids and monolayer cortical neurons to investigate infection of brain with pseudotyped SARS-CoV-2 viral particles. Spike-containing SARS-CoV-2 pseudovirus infected neural layers within brain organoids. The expression of ACE2, a host cell receptor for SARS-CoV-2, was sustained during the development of brain organoids, especially in the somas of mature neurons, while remaining rare in neural stem cells. However, pseudotyped SARS-CoV-2 was observed in the axon of neurons, which lack ACE2. Neural infectivity of SARS-CoV-2 pseudovirus did not increase in proportion to viral load, but only 10% of neurons were infected. Our findings demonstrate that brain organoids provide a useful model for investigating SARS-CoV-2 entry into the human brain and elucidating the susceptibility of the brain to SARS-CoV-2.

## 1. Introduction

The worldwide pandemic of the coronavirus disease 2019 (COVID-19) caused by SARS-CoV-2 infection become prolonged because of its extensive impact on multiple body organs and its complicated infection mechanism. A severe and acute respiratory illness accompanied by a fever were initially reported to be the major symptoms caused by infection with SARS-CoV-2 [1]. However, recent studies have identified other organ cells targeted by SARS-CoV-2 besides the cells of the respiratory system, such as intestinal cells [2], pancreatic cells [3], and cardiomyocytes [4]. Moreover, nervous system abnormalities, including impaired smell and taste, headache, and seizure have been reported [5]. In April 2020, the first case of meningitis related to SARS-CoV-2 was reported [6]. The specific RNA of SARS-CoV-2 was detected in the cerebrospinal fluid of the patient [6]. However, the mechanism of SARS-CoV-2 infection and susceptibility of the central nervous system is still elusive. 

Human pluripotent stem cell (hPSC)-derived brain organoids have been utilized for the modeling of human-specific viral infections of the brain. For example, the infection of hPSC-derived neurospheres or brain organoids with Zika virus (ZIKV) or cytomegalovirus (CMV) remodeled the clinical phenotypes such as microcephaly and impaired mature neurons [7,8,9,10]. However, there is very limited evidence of SARS-CoV-2 infection in the brain owing to the lack of experimental models that can recapitulate a SARS-CoV-2-infected brain. Consequently, there have been no studies on the mechanism of neuronal infection and treatment of brain symptoms related to SARS-CoV-2 infection.

Here we present a platform to explore the neural entry of pseudotyped SARS-CoV-2 by incubating hPSC-derived dorsal forebrain organoids with spike-containing SARS-CoV-2 pseudo-entry virus. We also utilized a monolayer culture of 2D cortical neurons as well as organoids to evaluate ACE2 expression and the infectivity of the SARS-CoV-2 pseudovirus. Our results show that mature neurons express ACE2 at the soma and are susceptible to the entry of the spike-pseudotyped SARS-CoV-2, establishing an in vitro model to study the impacts of SARS-CoV-2 on the human brain.

## 2. Materials and Methods

### 2.1. Antibodies

Anti-ACE2 (Santa Cruz Biotechnology, SC-390851, Dallas, TX, USA), anti-SATB2 (Abcam, ab51502, Eugene, OR, USA), anti-beta III Tubulin (TUJ-1) (Abcam, ab14545), anti-MAP2 (EMD Millipore, AB2290, Burlington, MA, USA), Anti-LaminA/C (Cell Signaling Technology, 2032, Danvers, MA, USA), and anti-SOX2 (Abcam, ab59776) antibodies were used as primary antibodies for immunohistochemistry, immunocytochemistry, and immunoblot assays. Goat anti-Rabbit IgG (H+L), TRITC (Thermo Fisher Scientific, A16101, Waltham, MA, USA), and goat anti-Mouse IgG (H+L), Alexa Fluor 488 (Thermo Fisher Scientific, A11001) were used as secondary antibodies for immunohistochemistry and immunocytochemistry. Goat anti-Rabbit IgG (H+L), HRP (Thermo Fisher Scientific, 31460) and goat anti-Mouse IgG (H+L), HRP (Thermo Fisher Scientific, 31430) were used as secondary antibodies for immunoblot assay.

### 2.2. Cell Culture

The human embryonic stem cell line H7 hESC was obtained from WiCell Research Institute (WA07, Madison, WI, USA) and 293T cells were purchased from the American Type Culture Collection (ATCC, CRL-3216, Manassas, VA, USA). H7 cells were grown on Matrigel-coated plates with TeSR-E8 medium (STEMCELL Technologies, Vancouver, BC, Canada). The 293T cells were maintained with Dulbecco’s Modified Eagle’s Medium (DMEM) containing 10% FBS and penicillin/streptomycin. All cells were maintained under a fully humidified atmosphere of 95% air and 5% CO_2_ at 37 °C.

### 2.3. Generation and Infection with SARS-CoV-2 Pseudovirus

Spike-containing SARS-CoV-2 pseudovirus was generated as previously described [11]. Briefly, 293T cells grown to 70% confluency were transfected with pcDNA3.1-SARS2-Spike (Addgene, 145032, Watertown, MA, USA), psPAX2 (Addgene, 12260), and pUltra-hot (Addgene, 24130) plasmids using Lipofectamine 3000 reagent (Thermo Fisher Scientific). After 6 h, the medium was replaced with fresh medium. After 48 h, the medium was collected and concentrated with PEG-it^TM^ virus precipitation solution (System Biosciences, Palo Alto, CA, USA). The MOI was calculated using the Lenti-X^TM^ qRT-PCR Titration Kit (Takara Bio, Kusatsu, Japan) according to the manufacturer’s manual.

### 2.4. Brain Organoid Differentiation

Dorsal forebrain organoid was generated from H7 hESCs adopting a previously reported protocol [12]. On day 0, H7 hESCs grown to 80–90% confluency were dissociated with Accutase (Gibco) and reaggregated in ultra-low-cell adhesion 96 well V-bottom plates (10,000 cells per well) with 100 mL of cortical differentiation medium (CDM) I (Glasgow-MEM containing 20% KSR, 0.1 mM MEM-NEAA, 1mM sodium pyruvate, 0.1 mM β-mercaptoethanol, and P/S) supplemented with 20 μM Y-27632 (day 0–6), 3 μM IWR-1 (day 0–18), and 5 μM SB431542 (day 0–18). The media was changed every 3 days until day 18. From day 18, the aggregates were moved to 60 mm ultra-low-attachment culture dishes on the orbital shaker (70 rpm) in CDM II (DMEM/F12 containing 2 mM glutamax, 1% N2, 1% CD lipid concentrate, 0.25 μg/mL fungizone, and P/S). From day 35, the organoids were incubated in CDM III (CDM II with 10% FBS, 5 μg/mL heparin, and 1% Matrigel). From day 70, the organoids were incubated in CDM IV (CDM III with B27 and 2% Matrigel). CDM II, CDM III, and CDM IV were changed every 2–3 days.

### 2.5. Monolayer Induction of the Cortical Neuron

Directed differentiation of hESCs to cortical neurons was carried out using a well-established protocol [13]. Briefly, the day before the start of the differentiation, H7 cells were passaged with 0.5 mM EDTA and plated at high density. The neuronal induction with neural induction medium was started when the cells reached 100% confluence (day 0). The medium was changed daily to the neural induction medium. On day 8, the neuroepithelial sheet was carefully lifted off using dispase, broken up gently into aggregates of 300–500 cells, and plated on poly-L-ornithine and laminin-coated plates in neuronal induction medium. On the next day (day 9), the media was changed to neuronal maintenance medium containing 20 ng/mL fibroblast growth factor 2 (FGF2). The medium was changed every other day and FGF2 was removed from the medium on day 13. Upon the appearance of neuronal rosette, the cells were split with dispase at a 1:2 ratio. On day 25, the cells were dissociated with Accutase and plated at a 1:1 ratio on poly-L-ornithine and laminin-coated plates. The cells were then expanded at 1:2 when they reached 90% confluence until day 30. The neural maintenance medium (1 L) consisted of 500 mL DMEM:F12 + glutamax (Thermo Fisher Scientific), 0.25 mL Insulin (10 mg/mL, Sigma, St. Louis, MO, USA), 1 mL β-mercaptoethanol (50mM Thermo Fisher Scientific), 5 mL non-essential amino acids (100 X Thermo Fischer Scientific), 5 mL sodium pyruvate (100 mM, Sigma), 2.5 mL Pens/Strep (10000 U/μL, ThermoFisher Scientific), 5 mL N2 (ThermoFisher Scientific), 10 mL B27 (Thermo Fisher Scientific), 5 mL glutamax (100 X, Thermo Fisher Scientific), and 500 mL Neurobasal (Thermo Fisher Scientific) medium. The neural induction medium consisted of neuronal maintenance medium, 1 μM dorsomorphin, and 10 μM SB431542.

### 2.6. Immunohistochemistry

The organoids were fixed with 4% formaldehyde in PBS for 20 min, washed in PBS twice, and equilibrated in 30% sucrose in PBS overnight at 4 °C. Then, the organoids were embedded in optimum cutting temperature (OCT) compound (Tissue Tek) and cryosectioned to 16 μm thickness. The sections were washed with 0.1% Triton X-100 in PBS and blocked with 5% donkey serum and 0.1% Triton X-100 in PBS at room temperature for 1 h. The sections were then incubated with primary antibodies (diluted with 2.5% donkey serum and 0.1% Triton X-100 in PBS to 1:500) overnight at 4 °C, followed by incubation with secondary antibodies (diluted to 1:600) at room temperature for 1 h. After three washes (10 min each) with PBS, the sections were incubated with diluted DAPI (1:1000) at room temperature for 20 min. Immunofluorescence images were taken with the Cytation^TM^ 5 Cell Imaging Multi-Mode Reader (BioTek, Winooski, VT, USA).

### 2.7. Immunocytochemistry

Immunocytochemistry of the monolayer-cultured cortical neurons was performed as previously described [14]. The cortical neurons were fixed with 4% formaldehyde in PBS for 15 min and washed in PBS twice, followed by permeabilization with 0.1% Triton X-100 in PBS for 15 min. The cells were blocked in 0.1% bovine serum albumin (BSA) in PBS at room temperature for 1 h and incubated with primary antibodies diluted (with 0.1% BSA in PBS to 1:500) overnight at 4 °C, followed by incubation with secondary antibodies diluted (with 0.1% BSA in PBS to 1:600) at room temperature for 1 h. After three washes (10 min each) with PBS, the cells were incubated with DAPI (1:1000) at room temperature for 15 min. Immunofluorescence images were taken with the Cytation^TM^ 5 Cell Imaging Multi-Mode Reader (BioTek).

### 2.8. Immunoblotting

Immunoblotting was performed as previously described [15]. For the extraction of proteins, the organoids were lysed with a PRO-PREP protein extraction solution (Intron), followed by centrifuge at 13,000 rpm for 20 min. Each protein sample was subjected to SDS-polyacrylamide gel electrophoresis (PAGE) for size-separation. Then, the separated proteins were transferred to polyvinylidene difluoride (PVDF) membranes using the semi-dry transfer (Bio-Rad). The membranes were blocked with skim milk and incubated with primary antibodies overnight at 4 °C, followed by incubation with HRP-conjugated secondary antibodies for 1 h at room temperature. The signals were detected using a chemiluminescence reagent (Abclon Inc., Seoul, Korea).

### 2.9. Statistical Analysis

Statistical significance of data was analyzed using the Student’s T-test (two-tailed) with Microsoft Excel 2019(Redmond, WA, USA). Data are presented as mean ± standard deviation (SD). Statistical differences were determined based on P-values (* *p* < 0.05, ** *p* < 0.01, and *** *p* < 0.001).

## 3. Results

### 3.1. Pseudotyped SARS-CoV-2 Infection of Brain Organoids that Express ACE2

To examine the entry of SARS-CoV-2 into neurons, we generated dorsal forebrain organoids and monolayer-cultured cortical neurons from human embryonic stem cells (hESCs). Spike-pseudotyped viral particles were generated through the lentiviral production system by utilizing SARS-CoV-2 spike-encoding plasmids instead of lentiviral envelope plasmids [16]. Next, 293T cells were co-transfected with three plasmids for the production of pseudotyped particles of SARS-CoV-2; the first plasmid is a lentiviral packaging vector containing core genes gag and pol but lacks an envelope gene; the second plasmid encodes SARS-CoV-2 spike protein; the third plasmid, a transfer vector, contains mCherry reporter gene and lentiviral long terminal repeat (LTR) that enables its integration into the host cell genome. Next, we treated brain organoids and 2D cortical neurons with the pseudotyped SARS-CoV-2 viral particles at a multiplicity of infection (MOI) of 20 (Figure 1A). First, brain organoids that had differentiated for 6 months were infected with the SARS-CoV-2 pseudovirus with mock-treatment as a control group (Figure 1B). After 48 h of incubation, the organoids were sectioned and stained with antibodies to a neural lineage marker, TUJ1. As shown in Figure 1C, the pseudo-SARS-CoV-2-infected cells were detected as mCherry-positive cells. The observation of mCherry/TUJ1-double-positive cells indicated that the neuronal lineage cells within the mature brain organoids are susceptible to the pseudo-SARS-CoV-2 infection (Figure 1B).

Given that SARS-CoV-2 expresses spike protein as a surface glycoprotein, we assumed that the spike of the SARS-CoV-2 pseudovirus would target ACE2-expressing cells in the brain organoids. Therefore, we stained mock-treated or pseudo-SARS-CoV-2-treated organoids with ACE2. Since an HIV-1-based lentiviral expression vector used for the generation of pseudovirus contains the coding sequence for mCherry, cells infected with pseudo-SARS-CoV-2 can be detected with mCherry signal. The mCherry signals from the pseudovirus-infected cells were co-localized with ACE2 (Figure 1C), indicating that the spike surface protein of SARS-CoV-2 can target ACE2-expressing cells in the brain organoids. The infectivity was approximately 10%, which was significantly higher than that of the mock-treated group, as calculated with the percentage of mCherry-positive cells among DAPI-positive cells (Figure 1D).

### 3.2. Sustained Expression of ACE2 during the Development of Brain Organoids

As ACE2 has been considered as a major receptor for the entry of coronaviruses, being recognized by spike proteins of SARS-CoV and SARS-CoV-2 [17,18], we monitored the protein level of ACE2 during the development of brain organoids at weekly intervals (Figure 2A). In our hands, a translucent neuroepithelium, which contains neural stem cells that can differentiate into neurons and glia, was observed within 2 weeks of differentiation (Appendix A). Western blot analysis showed that TUJ1, a marker for neural lineage cells, was expressed after 3 weeks, whereas SATB2, a marker for mature neurons, was detected at the late stage (Figure 2A). Interestingly, the expression level of ACE2 was sustained during the whole period of brain organoid development (Figure 2A).

Next, we performed immunohistochemistry analysis of sectioned organoids for the region-specific detection of diverse cell type markers as well as ACE2. The lamination of the progenitor zone (SOX2-positive layer) and neural layer (HOPX-positive layer) was incomplete at week 5 (Appendix A, upper). At later stages, mature neurons expressing CTIP2 or SATB2, but not SOX2, were observed in the outside layer, which is distinct from the progenitor zones such as the SOX2-positive subventricular and TBR2-positive intermediate zones (Appendix A, middle and bottom). Co-staining with ACE2 and the mature neuron marker, MAP2, showed that ACE2-positive cells were enriched in the mature neuron layer (Figure 2B). With the expansion of neuronal population at week 10, the distribution area of ACE2 was also enlarged (Figure 2B). These results suggest that the expression of ACE2 is sustainable during human brain development and was enriched in the mature neuron layer of the brain.

### 3.3. ACE2 Is Expressed in the Somas of Mature Neurons, but Rarely in the Progenitor Cells

To investigate whether ACE2 is expressed in the neural stem cells as well as in mature neurons, we co-stained brain organoids with ACE2 and SOX2. At the early stage of development (week 5) when the neural layer was not completely separated from the progenitor zones (Appendix A, upper), ACE2/SOX2-double positive cells were observed in the peripheral region of the organoids (Figure 3A, left). However, mature brain organoids (weeks 7 and 10) exhibited distinct localization of ACE2 and SOX2 (Figure 3A, middle and right). Considering that the late-stage organoids displayed a separated neuron layer from the progenitor zones (Appendix A, middle and bottom), these data indicate that neural stem cells hardly express ACE2 compared to the mature neurons.

We also assessed the expression pattern of ACE2 in 2D cultured cortical neurons. The pan-neuron marker, MAP2, was detected in both the somas and axons of mature cortical neurons (Figure 3B). Intriguingly, the ACE2 signal was clearly observed in the somas of the neurons, but was rarely detected at the axon (Figure 3B). Taken together, these data suggest that ACE2 is expressed in the somas of differentiated neurons, rather than in undifferentiated neural stem cells.

### 3.4. Neural Infectivity of Pseudotyped SARS-CoV-2 Is Independent of the Viral Load

To determine whether the infectivity of the pseudo-SARS-CoV-2 on neurons is dependent on the amounts of viral particles, we treated 2D cortical neurons with different doses (MOI = 1, 5, 20) of pseudotyped SARS-CoV-2. The mock-treated cortical neurons showed only TUJ1-single positive neurons (Figure 4A), whereas the pseudo-SARS-CoV-2-infected neurons exhibited a weak but clear mCherry signal, which indicates the infection by the viral particles (Figure 4B–D). The magnified images presented that mCherry signal was detected in the axon of the neurons (Figure 4B–D), despite the absence of ACE2 in the neural axon (Figure 3B). Quantification of the infected cell portion showed that the infectivity of pseudo-SARS-CoV-2 on neurons was not elevated in proportion to the increase in viral loads, but remained at approximately 10% (Figure 4E). These results imply that limited portions of the cortical neurons are susceptible to infection with the SARS-CoV-2 pseudovirus.

## 4. Discussion

Accumulated knowledge on the infection mechanism of SARS-CoV-2 has provided various strategies to prevent infection [19]. Unfortunately, lack of in vitro and in vivo models has made it difficult to capture the clinical condition of SARS-CoV-2 viral entry into actual human organs. In this present study, we successfully modeled the neural entry of SARS-CoV-2 pseudovirus using mature dorsal forebrain organoids from human embryonic stem cells. We cultured dorsal forebrain organoids up to six months, utilizing the protocol developed by the Sasai group [12], which recapitulates actual human brain-specific features like cortical development and distinct gene expression patterns [20]. Among the several distinct protocols for producing 3D brain organoids, the protocol we adopted here is known to generate the most consistent shape and transcriptomic signatures [21]. Thus, the brain organoid model of pseudo-SARS-CoV-2 entry we generated here can provide a useful disease model platform for high-throughput drug screening.

Based on the mammalian tissue expression databases showing the distribution of neurological tissue expression of ACE2, SARS-CoV-2 is considered to target the brain through interaction with neural ACE2 [22]. Here, we clarified the expression of ACE2 in both brain organoids and 2D cortical neurons. Previous studies showed that SARS-CoV-2 can enter into brainspheres that are 3D spheroids of differentiated neurons [23,24]. In addition to a culture of one neuronal cell type, we here employed an organoid model which contains mixed cell types with structural boundaries. We observed that ACE2 expression was sustained during brain development but was hardly detected in the progenitor regions. Indeed, the infectivity of pseudo-SARS-CoV-2 was similar in the 2D cortical neurons and brain organoids, a mixture of progenitor cells and mature cortical neurons.

We also found that the neural axon did not express ACE2, which suggests that ACE2-mediated entry of pseudo-SARS-CoV-2 did not occur in the synaptic terminals. Given a previous report showing that SARS-CoV-2 can extend into the neurites [23], the viral migration between neurons will be very active owing to the complex neural circuit. However, a recent study posted in bioRxiv reported that SARS-CoV-2 infection in neurons does not productively replicate [25]. In addition, our data showed that the infection rate of SARS-CoV-2 pseudovirus did not increase dose-dependently by increasing the viral load. This is thought to be due to the low neural infection rate of SARS-CoV-2 pseudovirus, approximately 10%, shown in our current study with a pseudovirus and another study using SARS-CoV-2 particles [25]. A recent study evaluated ACE2 expression across various cell types and organoids from human stem cells and demonstrated that cortical neurons expressed relatively low levels of ACE2 compared to other cells such as pancreatic endocrine cells, endothelial cells, cardiomyocytes, microglia, and dopaminergic neurons [3]. Based on these results, the degree of nerve infection will not be as severe as in other organs like the lungs, which are considered to be a major target organ. Nevertheless, in the case of the central nervous system, even a mild infection can lead to life-threatening consequences. Additionally, the possibility of the use of another unknown receptor for the entry of SARS-CoV-2 should not be excluded. Therefore, tremendous efforts should be put into finding ways to prevent brain infection with SARS-CoV-2.

Lastly, it should be noted that we only checked the entry of spike pseudovirus into neurons in the brain organoids, but did not consider the penetration of SARS-CoV-2 through the blood–brain barrier (BBB). To accurately model the SARS-CoV-2-infected brain, the permeability of the BBB to SARS-CoV-2 should be recapitulated in vitro. Future studies combining BBB models and brain organoids would estimate the impact of SARS-CoV-2 on the central nervous system, thus getting closer to clinical conditions.

In this study, we used a human embryonic stem cell line to produce brain organoids of a SARS-CoV-2 pseudovirus infection model. Induced pluripotent stem cells (iPSCs) from patients can be utilized as starting cells for dorsal forebrain organoids with high reproducibility [21]. Hence, customized modeling and prediction of SARS-CoV-2 infection to the brain would be possible by applying patient-derived iPSCs to our platform.

## Figures and Tables

**Figure 1 viruses-12-01004-f001:**
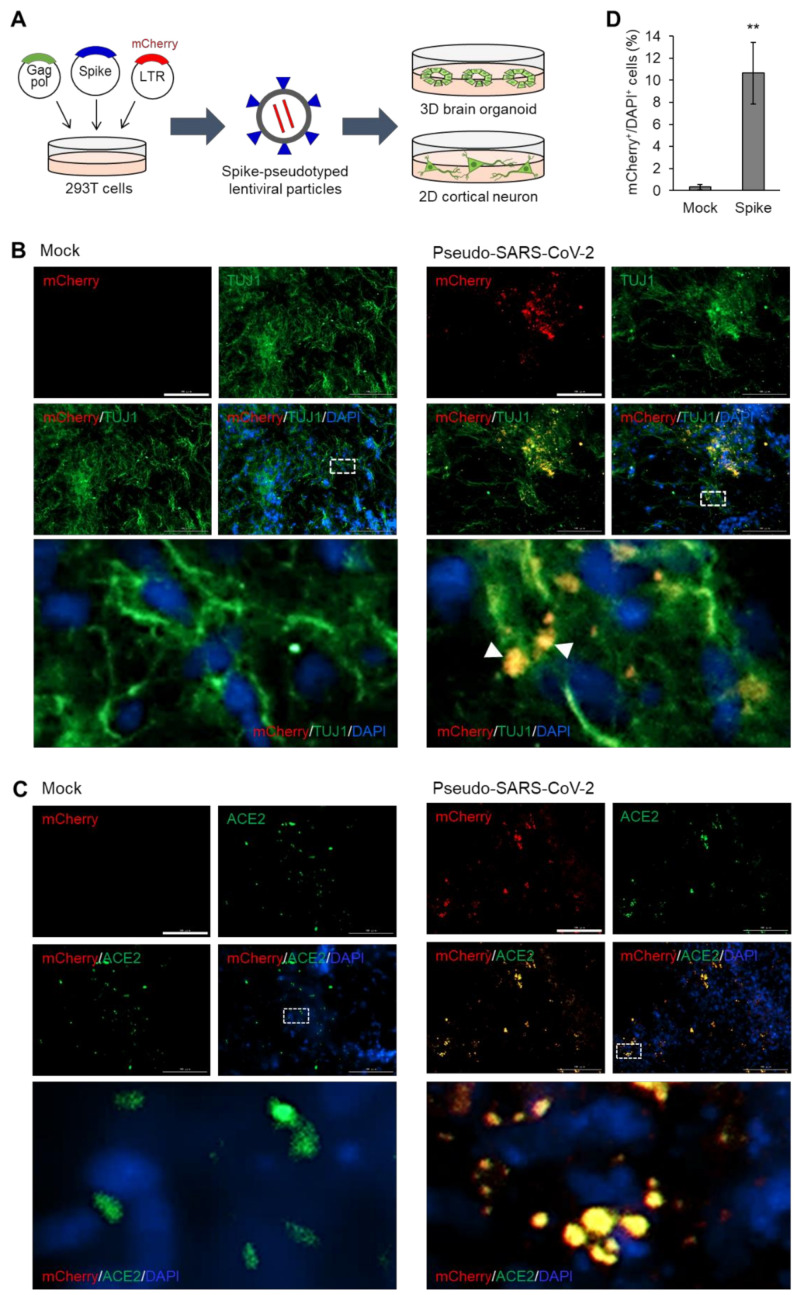
Infection of brain organoids with pseudotyped SARS-CoV-2. (**A**) Schematic illustration of pseudotyped SARS-CoV-2 infection of brain organoids and cortical neurons. (**B**) Immunofluorescence of mock-treated or SARS-CoV-2 pseudovirus-treated dorsal forebrain organoids (6 months) for detecting pseudotyped SARS-CoV-2 spike (mCherry) and TUJ1. (**C**) Immunofluorescence of mock-treated or SARS-CoV-2 pseudovirus-treated dorsal forebrain organoids (6 months) for detecting pseudotyped SARS-CoV-2 spike (mCherry) and ACE2. (**D**) Quantification of immunofluorescence indicates the percentage of mCherry-positive cells among the total DAPI-positive cells. In D, data are presented as mean ± SD (*n* = 4). * *p* < 0.05; ** *p* < 0.01; *** *p* < 0.001. Scale bar, 100 μm.

**Figure 2 viruses-12-01004-f002:**
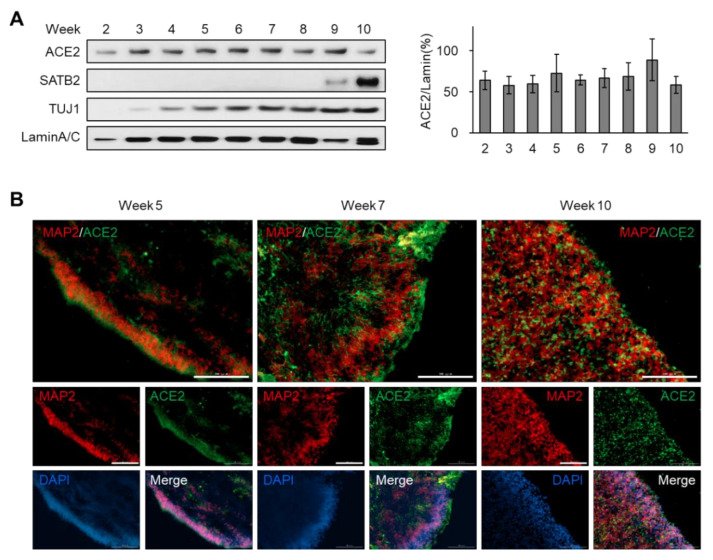
Sustained expression of ACE2 during the development of brain organoids. (**A**) Immunoblot analysis of dorsal forebrain organoids collected at different timepoints after differentiation. The relative intensity of the ACE2 band was quantified using LaminA/C as a control. (**B**) Immunofluorescence of dorsal forebrain organoids collected at different timepoints after differentiation for detecting MAP2 and ACE2. Scale bar, 100 μm.

**Figure 3 viruses-12-01004-f003:**
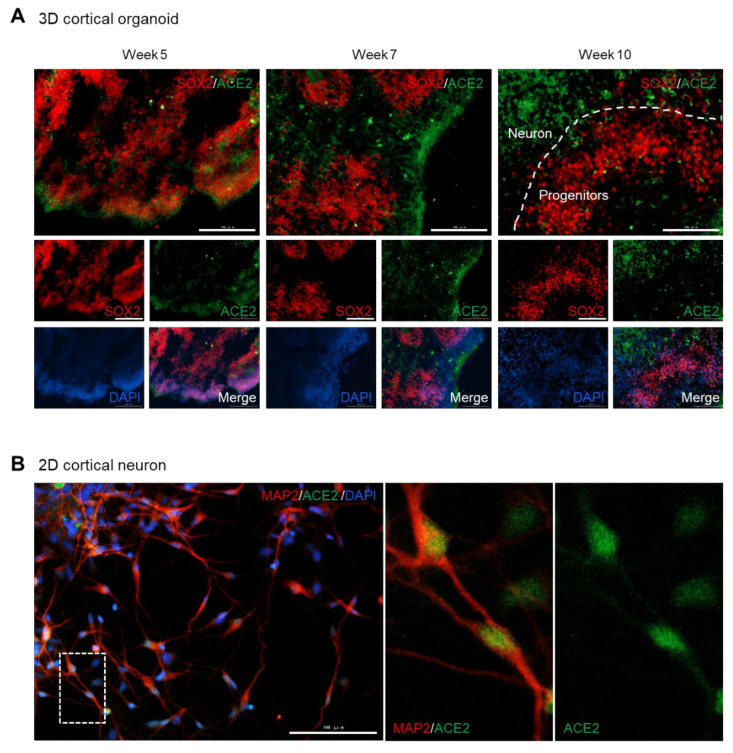
Somas of mature neurons, but not progenitor cells, express ACE2. (**A**) Immunofluorescence of dorsal forebrain organoids collected at different timepoints after differentiation for detecting SOX2 and ACE2. (**B**) Immunofluorescence of monolayer cultured cortical neurons collected 38 days after differentiation for detecting MAP2 and ACE2. Scale bar, 100 μm.

**Figure 4 viruses-12-01004-f004:**
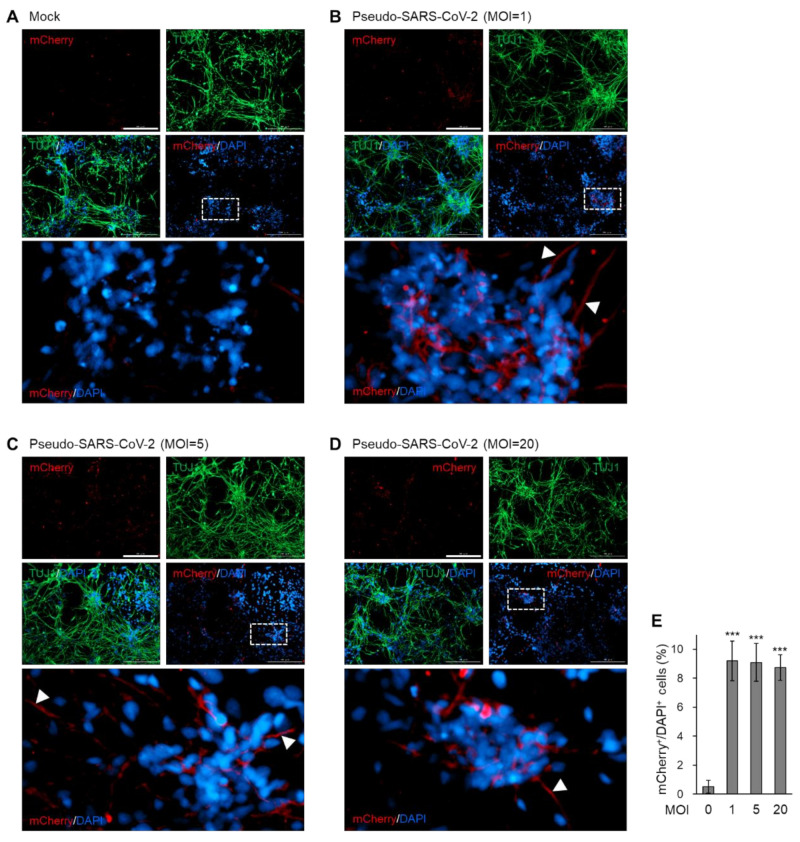
Neural infectivity of pseudotyped SARS-CoV-2 with different doses. (**A**) Immunofluorescence of mock-treated monolayer cortical neurons for detecting pseudotyped SARS-CoV-2 spike (mCherry) and TUJ1. (**B–D**) Immunofluorescence of monolayer cortical neurons, which were incubated with MOI = 1 (**B**), 5 (**C**), or 20 (**D**) of SARS-CoV-2 pseudovirus for detecting pseudotyped SARS-CoV-2 spike (mCherry) and TUJ1. White arrow, mCherry signal detected in axons. Scale bar, 200 μm. (**E**) Quantification of immunofluorescence indicates the percentage of mCherry-positive cells among the total DAPI-positive cells. In E, data are presented as mean ± SD (*n* = 4). * *p* < 0.05; ** *p* < 0.01; *** *p* < 0.001.

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
