# Peer review of "Infection of Brain Organoids and 2D Cortical Neurons with SARS-CoV-2 Pseudovirus"

_viruses, 2020, doi:10.3390/v12091004_

Round 1
Reviewer 1 Report
This is a very well performed study and presented paper on the susceptibility of brain/neurons for SARS-CoV2 pseudovirion infection.
Major issue: I have a problem with the originality of data: Ramadi et al. (the "Düsseldorf-Group") have used wild-type virus isolates to infect brain organoids. So I do not fully understand the merit of your approach that seems to represent a less authentic model than the German one and mainly produces data that have already been reported by the Düsseldorf-group. However, I would be happy if you could convince me by better explaining the rationale of your study and novelty of gained insights.
Minor issues:
Please go through the text and correct "infection of pseudo-viruses" by "infection by" or "...with pseudo-viruses". Its not the virus that gets infected, but the virus, which infects, and I know that this is what you want to say...
# Always use SARS-CoV2 pseudo virus instead of SARS-CoV2 virus to make clear that you worked with the former.
# Lines 235-236: Could it be possible that pseudo virus enters the cells via binding to soma ACE-2 receptor and then migrate to axons? Could the use of alternative receptors be envisaged? This may be worth discussing.
Author Response
This is a very well performed study and presented paper on the susceptibility of brain/neurons for SARS-CoV2 pseudovirion infection.
We thank the reviewer for recognizing the potential importance of our findings. We feel our responses to his/her criticisms have improved the manuscript with several corrections. Our responses to the reviewer’s queries are described point-by-point below.
Major issue: I have a problem with the originality of data: Ramadi et al. (the "Düsseldorf-Group") have used wild-type virus isolates to infect brain organoids. So I do not fully understand the merit of your approach that seems to represent a less authentic model than the German one and mainly produces data that have already been reported by the Düsseldorf-group. However, I would be happy if you could convince me by better explaining the rationale of your study and novelty of gained insights.
We used pseudotyped SARS-CoV-2 viral particles instead of live SARS-CoV-2 for two reasons:
(1) We wanted to measure the pure infection rate by excluding replication of virus in host cells. After infection, the genomic RNA of real SARS-CoV-2 is released into the cytoplasm of host cells, and translation of the viral RNA can produce structural viral proteins. Then, virus assembly occurs within host cells, followed by release and further infection of neighboring cells. In contrast, SARS-CoV-2 pseudovirus cannot go through this cycle, thereby enable us to estimate the initial infection rate upon exposure to spike-containing SARS-CoV-2, which allowed us to gain insights into the accurate susceptibility of neurons to SARS-CoV-2.
(2) The second reason is more down to earth. Due to the high pathogenicity and infectivity, live SARS-CoV-2 has to be handled only at authorized facilities. However, it takes too long to fulfill the strict biosafety level and get permission from the authorities. Despite this limitation, we wanted to study the susceptibility of brain to SARS-CoV-2 using human brain organoids and we got important findings using the pseudovirus and organoids. We felt a kind of responsibility to report this discovery quickly in this worldwide pandemic situation.
Minor issues:
Please go through the text and correct "infection of pseudo-viruses" by "infection by" or "...with pseudo-viruses". Its not the virus that gets infected, but the virus, which infects, and I know that this is what you want to say...
We corrected the text “infection of pseudo-viruses” to “infection by” or “infection with” as the reviewer pointed out (lines 77 and 242 in revised manuscript).
# Always use SARS-CoV2 pseudo virus instead of SARS-CoV2 virus to make clear that you worked with the former.
We clarified the use of “pseudovirus” in all parts demonstrating our present work, as the reviewer mentioned (lines 29, 53, 185, 187, 245, 247, 250, 253, 261, 267, 279, 285, and 302 in revised manuscript).
# Lines 235-236: Could it be possible that pseudo virus enters the cells via binding to soma ACE-2 receptor and then migrate to axons? Could the use of alternative receptors be envisaged? This may be worth discussing.
mCherry signal does not mean the presence of pseudovirus, because we used pseudovirus that contains lentiviral transfer vector with LTRs instead of SARS-CoV-2 RNA. The genes contained in transfer vector are integrated into the host cell genome. Therefore, the pseudo-SARS-CoV-2-infected cells express mCherry gene which were transduced from the lentiviral transfer vector. So, we cannot determine whether the pseudovirus enters into axons through other receptors or they migrate to axons from soma. We can only say that the integrated mCherry sequence by the SARS-CoV-2 pseudovirus are expressed in the neurons and the mCherry signal can be detected also in axons. With respect to the possibility of using other receptors, we mentioned it in discussion part (lines 292-293 in revised manuscript).

Reviewer 2 Report
This manuscript is fairly straight forward and describes the infection of brain organoids by SARSCoV-2. Observations relating to the ACE2 receptor during development and infectivity of neuronal cells are interesting, and of heightened importance due to the current COVID-19 pandemic.
Unfortunately the manuscript needs significant editing and at times this involves the description of experiments. Since the experiments are conceptually quite simple it was possible to use the Figure to interpret the findings, but this is obviously not ideal.
I have listed a few grammatical corrections.

Author Response
This manuscript is fairly straight forward and describes the infection of brain organoids by SARSCoV-2. Observations relating to the ACE2 receptor during development and infectivity of neuronal cells are interesting, and of heightened importance due to the current COVID-19 pandemic.
We thank the reviewer for recognizing the potential importance of our findings. We feel our responses to his/her criticisms have improved the manuscript with several corrections. Our responses to the reviewer’s queries are described point-by-point below.
Unfortunately the manuscript needs significant editing and at times this involves the description of experiments. Since the experiments are conceptually quite simple it was possible to use the Figure to interpret the findings, but this is obviously not ideal.
We agree that the short description and simple figure are not sufficient for understanding of our experiments. In particular, more detailed description is needed for the production of pseudotyped SARS-CoV-2 viral particles. In revised version of our manuscript, we accurately demonstrated the three plasmids utilized in this study to produce the pseudovirus (lines 158-163 in revised manuscript).
I have listed a few grammatical corrections.
We thank the referee for the way to improve our manuscript. We corrected the points as below.
- Suggested rewrite of the abstract. Abstract: Since the global outbreak of SARS-CoV-2 (COVID-19), infections of diverse human organs along with multiple symptoms continue to be reported. However, the susceptibility of the brain to SARS-CoV-2, and mechanisms underlying neurological infection are still elusive. Here, we utilized human embryonic stem cell-derived brain organoids and monolayer cortical neurons to investigate infection of brain with pseudotyped SARS-CoV-2 viral particles. Spike-containing SARS-CoV-2 pseudovirus infected the neural layer within brain organoids. The expression of ACE2, a host cell receptor for SARS-CoV-2, was sustained during the development of brain organoids, especially in the somas of mature neurons, while remaining rare in neural stem cells. However, pseudotyped SARS-CoV-2 was observed in the axon of neurons, which lack ACE2. Neural infectivity of SARS-CoV-2 did not increase in proportion to viral load, and only 10% of neurons were infected. Our findings demonstrate that brain organoids provide a useful model for investigating SARS-CoV-2 entry into the human brain and elucidating the susceptibility of the brain to SARS-CoV-2.
We have updated the abstract as the reviewer recommended.
- Line 164. Should be “is” susceptible
It is not right, because the neuronal lineage cells “are” susceptible, not “is” susceptible. Thus, we suggest leaving it as it is.
- Line 154. ……..into neurons
We corrected the sentence as the reviewer pointed out.
- Line 161. With antibodies to
We corrected the sentence as the reviewer pointed out.
- Line166 Should be “Given that SARS-CoV-2 expresses spike protein as a surface glycoprotein..
We corrected the sentence as the reviewer pointed out.
Line 169. It was a little unclear whether antibodies were being used here.
The use of “detected” apparently confused the reviewer. We removed the word and explained the principle of detecting infected cells with mCherry signal in detail (lines 173-175 in revised manuscript).
Line 185. Suggested rewrite. As ACE2 has been considered as a major receptor for the entry of coronaviruses, being recognized by spike proteins of SARS-CoV and SARS-CoV-2 [17,18], we monitored the protein level of ACE2 during the development of brain organoids at weekly intervals (Figure 2A)
We corrected the sentence as the reviewer pointed out.
Line 187. The manuscript states……..In our hands, a translucent neuroepithelium was observed within 2 weeks of differentiation (Figure S1A). The authors should be more descriptive regarding the neuroepithelium
We agree with the reviewer’s opinion that neuroepithelium should be explained for the readers in broad fields. Hence, we added short description for the neuroepithelium (lines 194-195 in revised manuscript).
